# Peer review of "COVID-19 Vaccine Hesitancy in Italian Adults with Marfan Syndrome: Insights from a Secondary Analysis of a Cross-Sectional Study"

_vaccines, 2023, doi:10.3390/vaccines11040734_

Round 1

Reviewer 1 Report

Sample size was small. What was the rate of vaccine hesitancy in general population in Lombardy or other regions of Italy? What are the main reasons of vaccine hesitancy in Lombardy and in Italy?  Results of the report emphasized vaccine hesitancy in general population - MFS patients are just specific group of population.  

Author Response

Reviewer 1

Comment 1:

Sample size was small. What was the rate of vaccine hesitancy in general population in Lombardy or other regions of Italy?

Answer1:

Considering sample size, we added additional information in the methods as follow to clarify that in the case of MFS, 112 patients from Lombardy were adequate to reflect this regional population:

[…] The previously published study used a single-center and convenience sampling approach [20]. The authors used one proportion from a finite population approach to estimate a sample size that would represent patients with MFS living in Lombardy, Italy. To determine the appropriate sample size for the study, the researchers used a formula that accounted for the population size of roughly 1500 patients in Lombardy. The formula included a variable X, which was determined by a few factors: the sample proportion of 8%, which was the rate of mental health scores using the 12-item Short Form Survey under the third quartile of a previous study [17] and the margin of error set to 5%. The formula was X/(X+population size−1)=Zα/22*(1−sample proportion)/(margin of error). The required minimum sample size for a 95% confidence interval and a sample proportion of 8% (the rate of mental health patients under the 75th percentile) was, therefore, 106 patients. In other words, even if the sample of 112 enrolled patients in the analysis is limited, it was adequate to represent the population living in Lombardy based on this estimate. 

Comment 2:

What are the main reasons of vaccine hesitancy in Lombardy and in Italy?  Results of the report emphasized vaccine hesitancy in general population - MFS patients are just specific group of population. 

Answer2: Dear reviewer, thank you for your feedback. We have taken your comments into consideration and edited the discussion section of our manuscript accordingly. Specifically, we have focused on the main reasons for vaccine hesitancy in Lombardy and Italy. We have also added several analyses to provide a more detailed picture of the factors contributing to vaccine hesitancy in this population. We hope that these changes address your concerns and improve the overall quality of our manuscript. Thank you again for your valuable feedback.

Reviewer 2 Report

I was invited to revise the paper entitled "COVID-19 vaccine hesitancy in Italian adults with Marfan Syndrome: insights from a secondary analysis of a cross-sectional study". It was a brief report reporting the results of a sub-analysis of a cross-sectional study. It aimed to evaluate attitudes towards covid19 vaccination in a group of patients with Marfan Syndrome. 

Poor data were available in vaccine hesitancy among patients with MS and this study can improve the knowledge on this field.

My main concern about this report was the poor statistical analysis performed. Authors limited their analysis of differences in baseline characteriscts between hesitant and non-hesitant patients. I suggest to improve analysis performinfg regression model to evaluate factors associated to vaccine hesitancy. In addition, in table 1, Authors should perform a statistical correction for multiple comparisons.

Among discussion, Authors should discuss about differences in factors associated to vaccine hesitancy in MF patients compared to general population.

Author Response

Reviewer 2

Comment 1:

I was invited to revise the paper entitled "COVID-19 vaccine hesitancy in Italian adults with Marfan Syndrome: insights from a secondary analysis of a cross-sectional study". It was a brief report reporting the results of a sub-analysis of a cross-sectional study. It aimed to evaluate attitudes towards covid19 vaccination in a group of patients with Marfan Syndrome.

Poor data were available in vaccine hesitancy among patients with MS and this study can improve the knowledge on this field.

My main concern about this report was the poor statistical analysis performed. Authors limited their analysis of differences in baseline characteriscts between hesitant and non-hesitant patients. I suggest to improve analysis performinfg regression model to evaluate factors associated to vaccine hesitancy. In addition, in table 1, Authors should perform a statistical correction for multiple comparisons.

Answer 1:

Thank you for your feedback. We are glad to hear that you found our study valuable in improving the knowledge of vaccine hesitancy among patients with Marfan Syndrome. We took your comments into consideration and extensively added new analyses to provide a more detailed picture of vaccine hesitancy in this population. We hope our revised manuscript better addresses the research gap and provides useful insights to the scientific community and clinicians.

In relation to the advice to perform statistical correction for multiple comparisons in table 1, we agree that it is true that the probability of a false positive result increases with the number of statistical tests performed. However, in this case, only two comparisons per variable were made, which is a relatively small number of tests. Therefore, the risk of inflating the type I error rate is limited, and considering that we have reported only two significant differences (age and previous positive COVID-19 tests), the likelihood of having an inflated type 1 error is very limited. In other words, since only a small number of tests were conducted, the probability of a false positive result is not as high as it would be if a larger number of tests had been performed.

Comment 2:

Among discussion, Authors should discuss about differences in factors associated to vaccine hesitancy in MF patients compared to general population.

Answer2: Dear reviewer, thank you for your feedback. We have revised the discussion section, also concerning the new findings derived from the analyses you suggested. We appreciate your suggestions and hope that these changes have addressed your concerns.

Round 2

Reviewer 2 Report

Authors strongly improved the manuscript according to previous suggestions. About multiple comparisons, table 1 reports more than 20 statistical tests so it requires multiple comparisons correction.

Author Response

Dear Reviewer,

Thank you for your feedback and for highlighting the need for statistical corrections for the tests performed between hesitant and non-hesitant patients. We added to part in the methods (statistical analysis): […] In this analytical stage, as multiple comparisons were planned, the Bonferroni correction was applied to determine adequate probabilistic thresholding to mitigate the likelihood of false positive inferential results [25]. Therefore, the significance level for the multiple comparisons was α/k, where α was 0.05 and k (i.e., number of comparisons) was 21, implying an adjusted α equal to 0.002. In addition, we added a note in table one to highlight that significant differences require p < adjusted α (therefore, p < 0.002). Consistently, we revised some phrases in the results and discussion to provide clarity and caution in interpreting the comparisons of table 1. 
